# PreDicta chip-based high resolution diagnosis of rhinovirus-induced wheeze

Katarzyna Niespodziana[1], Katarina Stenberg-Hammar[2,3], Spyridon Megremis[4], Clarissa R. Cabauatan[1], Kamila Napora-Wijata[1], Phyllis C. Vacal [1], Daniela Gallerano[1], Christian Lupinek[1], Daniel Ebner[5], Thomas Schlederer[5], Christian Harwanegg[5], Cilla Söderhäll[3], Marianne van Hage[6], Gunilla Hedlin[2,3], Nikolaos G. Papadopoulos[4,7] & Rudolf Valenta[1]

Rhinovirus (RV) infections are major triggers of acute exacerbations of severe respiratory diseases such as pre-school wheeze, asthma and chronic obstructive pulmonary disease (COPD). The occurrence of numerous RV types is a major challenge for the identification of the culprit virus types and for the improvement of virus type-specific treatment strategies. Here, we develop a chip containing 130 different micro-arrayed RV proteins and peptides and demonstrate in a cohort of 120 pre-school children, most of whom had been hospitalized due to acute wheeze, that it is possible to determine the culprit RV species with a minute blood sample by serology. Importantly, we identify RV-A and RV-C species as giving rise to most severe respiratory symptoms. Thus, we have generated a chip for the serological identification of RV-induced respiratory illness which should be useful for the rational development of preventive and therapeutic strategies targeting the most important RV types.

[1] Division of Immunopathology, Department of Pathophysiology and Allergy Research, Center for Pathophysiology, Infectiology and Immunology, Medical University of Vienna, A-1090 Vienna, Austria. [2] Astrid Lindgren Children's Hospital, Karolinska University Hospital, SE-171 76 Stockholm, Sweden. [3] Department of Women's and Children's Health, Karolinska Institutet, SE-171 77 Stockholm, Sweden. [4] Division of Infection, Immunity & Respiratory Medicine, University of Manchester, Manchester M13 9NT, UK. [5] Phadia Austria GmbH, Part of Thermo Fisher Scientific ImmunoDiagnostics, A-1220 Vienna, Austria. [6] Immunology and Allergy Unit, Department of Medicine Solna, Karolinska Institutet and University Hospital, SE-171 77 Stockholm, Sweden. [7] Allergy Department, 2nd Pediatric Clinic, University of Athens, 106 79 Athens, Greece. These authors contributed equally: Katarina Stenberg-Hammar, Spyridon Megremis. Correspondence and requests for materials should be addressed to N.G.P. (email: ngpallergy@gmail.com) or to R.V. (email: rudolf.valenta@meduniwien.ac.at)

    1

Respiratory viral infections are among the most common triggers of acute exacerbations of pre-school wheeze, asthma, and chronic obstructive pulmonary disease (COPD)[1–3]. Asthma and COPD are severe and disabling diseases of the respiratory tract and hence represent a serious global health problem affecting different age groups. Acute pre-school wheeze and community-acquired pneumonia (CAP) are other common causes of emergency visits with possible viral etiology. There is an increasing prevalence of these airways diseases, rising treatment costs, and therefore virus-induced respiratory illnesses are a heavy burden for patients and the community[4,5]. Respiratory viral infections, mainly due to rhinovirus (RV), are responsible for approximately 80% of wheeze and asthma exacerbations in children[6,7]. Moreover, infants with rhinovirus-induced wheeze have a significantly increased risk for subsequent development of recurrent wheeze and childhood asthma[8]. Since exposure to RV does not lead to wheezing illness in all children, additional factors such as the host genotype, defects of the respiratory epithelial barrier, and/or atopic predisposition have been suggested to play important roles in asthma[9–11].

RV is genetically a highly diverse virus group with more than 160 distinct RV types which have been divided into three distinct RV species, RV-A, RV-B, and RV-C[12,13]. Rhinoviruses can also be classified according to which cellular receptor on human respiratory epithelial cells they use for entry[14]. RV-B and most RV-A variants bind to the intercellular adhesion molecule-1 (ICAM-1) (i.e., major RV group), while a subset of RV-A species binds to the low-density lipoprotein receptor (i.e., minor RV group)[15,16]. More recently, a cadherin-related family member 3 protein (CDHR3) has been reported as a one of the probable receptors for the RV-C species[17].

The identification of the culprit rhinovirus species responsible for severe exacerbations of respiratory disease is an extremely important topic as certain RV species (e.g., RV-C) are suspected to be associated with more severe wheezing illnesses and acute asthma exacerbations in infants and children compared to others[18,19]. In fact, there are also several preventive and therapeutic strategies for RV infections under development which require a precise knowledge of the clinically relevant RV species to be targeted. For example, several approaches for developing vaccines based on polyvalent inactivated RV, synthetic RV-derived peptides and recombinant RV proteins have been reported[20–25]. The formulation of a broadly protective vaccine obviously requires the inclusion of the clinically most relevant and common RV species. Furthermore, it has been shown that blocking of the viral receptor on respiratory epithelial cells (e.g., ICAM-1) can prevent RV infection[26]. Again therapeutic approaches targeting the viral receptors require knowledge which RV species are the most frequent and relevant ones. Finally, it is important to investigate the role of the different RV species for exacerbations of severe bronchial obstruction in different populations and for different age groups and manifestations of respiratory illness (e.g., pre-school wheeze, asthma, COPD, asthma-COPD overlap: ACO, CAP). While RV is well established as an important trigger factor for childhood wheeze and asthma, less is known regarding the role of RV infections in exacerbations of COPD and in respiratory disease exacerbations of older subjects[3]. Furthermore, the causal relationship of RV with CAP is also unknown[27].

Currently, the detection of RV in the course of respiratory infections is mainly based on reverse transcription of viral RNA and DNA amplification by polymerase chain reaction (PCR)[28]. Such tests can demonstrate the presence of virus-derived nucleic acid but they do not necessarily indicate that the particular virus had caused an infection and is indeed responsible for clinical symptoms in the patient[29]. In fact, rhinovirus RNA has been found in a high proportion of asymptomatic infants and children[30–32]. Furthermore, little is known about levels and epitope-specificities of natural antibody responses capable of neutralizing rhinoviruses, and thus protecting individuals against RV infections. Such information would be helpful for the development of new immunological strategies for the treatment and prevention of RV-induced exacerbations of respiratory diseases. Therefore, there is a huge and so far unmet need for high-resolution serological detection of rhinovirus infections. We have previously identified the capsid protein VP1 and an N-terminal VP1 peptide as major target for the natural antibody response of RV-infected subjects[33]. Then we have demonstrated that in vivo inoculation of subjects with RV16 indeed induced increases of VP1-specific antibody responses which were best detected with the VP1 protein of the corresponding species[33]. Similar increases of RV-specific antibodies were found in pre-school children after asthma attacks using complete recombinant RV capsid proteins[34]. In a recent study performed in pre-school children with acute asthma attacks, we observed that increases of RV-specific antibody responses reflected the severity of respiratory symptoms[35].

In the present study funded by the European Union project "PreDicta" (https://cordis.europa.eu/project/rcn/96868_en.html), we investigated if it is possible to generate a microarray-based serological test which can discriminate RV-A, RV-B, and RV-C as culprit species involved in childhood asthma attacks.

## Results

**Development of a high-resolution PreDicta microarray.** Figure 1 shows the arrangement of RV-derived proteins, peptides as well as control proteins on the PreDicta chip and the selection procedure for the N-terminal VP1 peptides. Recombinant capsid proteins (VP1-VP4) and fragments thereof representing RV-A, B, C species as well as non-structural proteins from RV89 were included (Supplementary Tables 1–3). Based on our previous finding that antibodies from RV-infected patients react preferentially with the N-terminus of VP1 (ref. [33]), we included synthetic N-terminal VP1 peptides from 30 RV strains which were selected in a rational, multistep process to represent distinct RV strains (Fig. 1a, Supplementary Table 2). Starting with 107 VP1 sequences retrieved from the NCBI database (RV-A: 76; RV-B: 25; RV-C: 6), multiple sequence alignments were performed to identify clusters of peptides with high degrees of sequence identities (Fig. 1a, Supplementary Table 2: Clusters A1–A17; B1–B9; C1–C3). In a next step, peptides were re-clustered into groups taking the chemical properties of amino acids into consideration (Supplementary Table 2: AI–AXVIII; BI–BIX; CI–CIII). From these groups 30 peptides with the most distantly related sequences were selected (Fig. 1b, c; Supplementary Tables 2 and 3). For a further refinement of VP1, VP2, and VP3 antibody responses, VP2 and VP3 fragments as well as peptides spanning the complete VP1 sequence from RV89 were included (Fig. 1d, Supplementary Tables 4 and 5). For control and calibration purposes we added recombinant allergens and control proteins (Fig. 1d, Supplementary Table 6) to the PreDicta chip.

Proteins and peptides were spotted in triplicates on a pre-activated glass slide containing six microarrays surrounded by a Teflon frame so that one chip allows testing of six serum samples (Fig. 1d). Identities and purities of recombinant proteins and peptides were tested by sodium dodecyl sulfate polyacrylamide gel electrophoresis (SDS-PAGE) followed by Coomassie Brilliant Blue staining, western-blotting, and by mass spectrometry, respectively (Supplementary Tables 3–5, Supplementary Fig. 1). Supplementary Table 7 shows that the PreDicta microarray allows reproducible measurement of IgG levels to the antigens according to intra- and inter-assay variations.

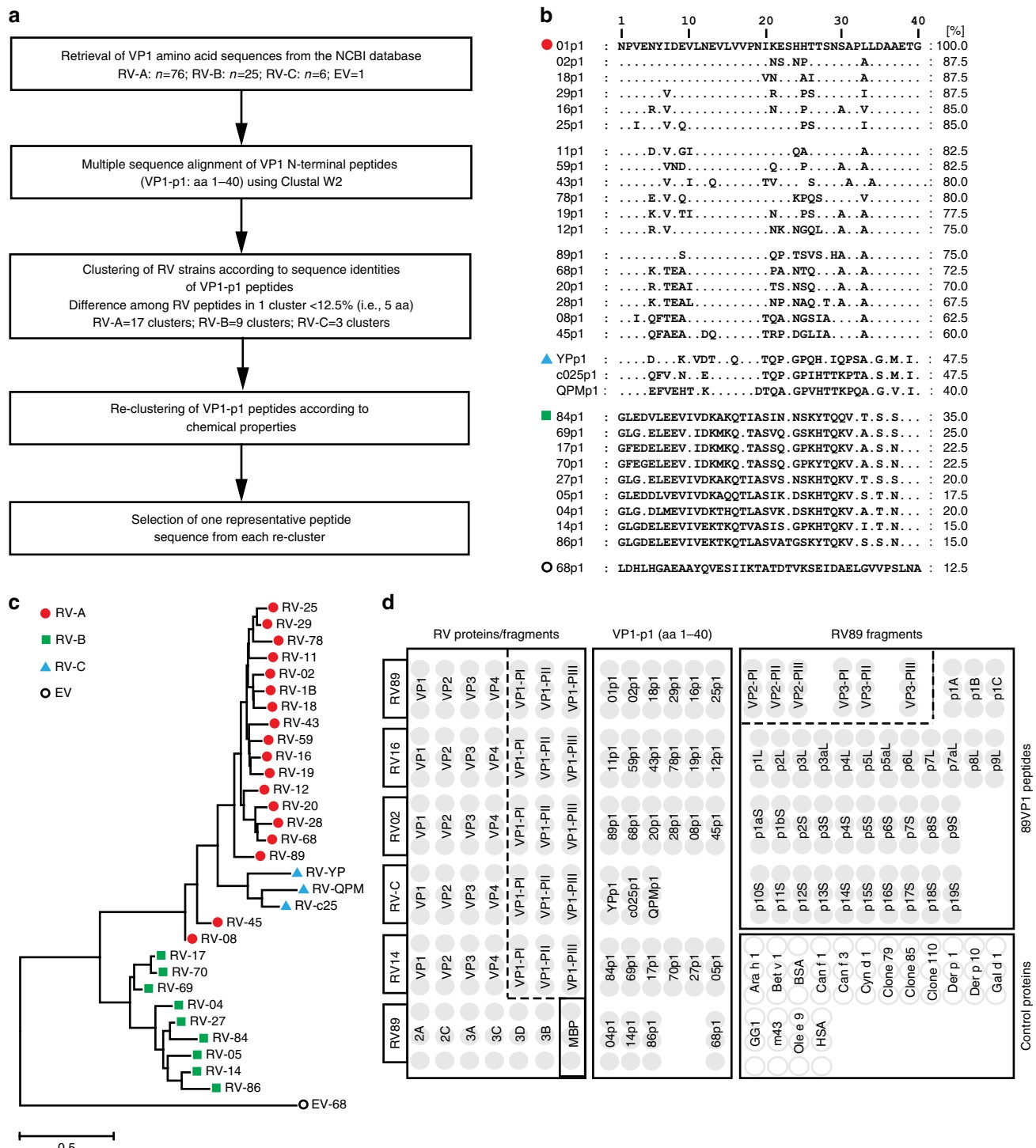

**Fig. 1** Composition of the PreDicta chip. **a** Selection process of VP1 peptides representing different RV strains as summarized in Supplementary Table 2. **b** Multiple sequence alignment of selected VP1 peptides. Left margin: Strain numbers (red circle: RV-A; blue triangle: RV-C, green square: RV-B, black circle: Enterovirus 68). Dots in the alignment represent identical amino acids. Sequence identities (%) with peptide 1 from RV 1 (01p1; top line) are shown for each peptide on the right margin. **c** Phylogenetic tree of the VP1 peptide sequences in **b**. **d** RV microarray layout depicting the positions of recombinant RV proteins/protein fragments, synthetic peptides (VP1-p1, 89VP1-derived peptides), MBP (maltose binding protein), and other control proteins which were spotted in triplicates. Proteins and peptides are designated as described in Supplementary Tables 3–6

**RV strain recognition is broader in older wheezing children.** The PreDicta chip was tested with sera from 120 pre-school children who were admitted to the hospital due to an acute wheezing episode[35]. Table 1 summarizes demographic and clinical data of the children investigated in this study. To investigate if the spectrum of recognized RV peptides varies by age, children

were grouped according to age (Group I: <1 year, $n = 35$; Group II: 1–2 years, $n = 53$; Group III: >2 years, $n = 32$). Figure 2a, b show the frequency and intensity of IgG and IgA antibody reactivity of the children to the 30 N-terminal VP1 peptides representing RV-A, RV-B, and RV-C species. The most frequent and highest IgG responses were directed to RV-C-derived

**Table 1 Demographic and clinical characterization of wheezing children**

| Characteristic | Group I(N=35) | Group II(N=53) | Group III(N=32) | Total(N=120) |
|---|---|---|---|---|
| Age (months) | | | | |
| Median | 10 | 17 | 32 | 18 |
| Range (Min–Max) | 6–12 | 13–24 | 25–42 | 6–42 |
| Gender | | | | |
| Male:female ratio | 26:9 | 31:22 | 19:12[a] | 76:43 |
| Male (%) | 74 | 58 | 61 | 63 |
| Ever wheeze before, n (%) | 23 (66) | 44 (83) | 25 (83)[a] | 92 (77) |
| Allergic sensitization[b] | | | | |
| Food allergens, n (%) | 5 (14) | 7 (13) | 10 (31) | 22 (18) |
| Respiratory allergens, n (%) | 1 (3) | 4 (7) | 4 (12.5) | 9 (7.5) |
| Hospitalized at the acute visit, n (%) | 26 (74) | 44 (83) | 26 (87)[a] | 96 (80) |
| Weeks until follow-up visit | | | | |
| Median | 11[a] | 11.5[a] | 12.5 | 11 |
| Range (Min–Max) | 7–27 | 7–24 | 9–30 | 7–30 |
| Presence:absence of cold symptoms | 35:0 | 50:3 | 30:2 | 115:5 |
| Days with respiratory symptoms (%) | | | | |
| Median | 13 | 11[a] | 6.5 | 11 |
| Range (Min–Max) | 1–85 | 0–100 | 1–63 | 0–100 |
| Days with $\beta_2$-agonists (%) | | | | |
| Median | 20 | 22[a] | 29.5 | 22 |
| Range (Min–Max) | 0–100 | 0–100 | 0–100 | 0–100 |

[a]1 or 2 values are missing for this variable
[b]Allergen-specific IgE to food and/or respiratory allergens determined by MEDALL allergen chip for the acute and follow-up visits (≥0.3 ISU-E was considered positive)

peptides followed by RV-A peptides whereas IgG reactivity to RV-B peptides was less frequent and intense (Fig. 2a). Similar results were obtained for IgA responses which in general were lower than the IgG responses (Fig. 2b). The analysis of IgG reactivity to structural and non-structural proteins and to recombinant fragments and synthetic peptides spanning VP1, VP2, and VP3 from RV89 is shown in Supplementary Fig. 2a for all 120 children and in Supplementary Fig. 2b for those children (n = 41) who had shown increases of RV89-specific antibody responses in follow-up serum samples taken after recovery. Results obtained thus confirmed our earlier observations showing that the majority of RV-specific antibody responses are directed against the N-terminus of VP1 as represented by the 30 peptides from the N-terminus of VP1 proteins from the different RV species[33]. Some children showed an IgG response against VP2 and in particular to a fragment representing the middle portion of VP2 whereas VP3 and VP4 showed no relevant IgG reactivity (Supplementary Fig. 2a, b). Among the non-structural proteins, 2C, 3A, and 3C showed some IgG reactivity (Supplementary Fig. 2).

The analysis of IgG reactivity according to the age of children at the acute visit showed that children <1 year of age had much lower RV-specific IgG levels compared to children who were older than 1 year. There was almost no change of the RV strain-specificity pattern among the three age groups (Fig. 2c). However, we found a positive correlation between the number of recognized VP1 peptides and the age of the children (Fig. 2d). Children between 6 and 12 months of age recognized a significantly lower number of N-terminal VP1 peptides than older (13–42 months) children. Children older than 2 years reacted with significantly more peptides than both groups of younger children (Fig. 2e).

The patterns of antibody response against different RV peptides were also analyzed using an independent bioinformatics approach (Supplementary Fig. 3). A phylogenetic clustering of all peptides was prepared and used to generate peptide groups according to sequence homology which represented to a large extent the RV subgroups A, B, and C. Then an unsupervised computer algorithm was used to cluster the patterns of antibody responses. Finally, the results of these analyses were super-imposed. It turned out that there was a very strong correlation between the two groupings, one based on antibody reactivity and the other based on sequence identities among peptides made through the unsupervised analysis. Thus antibody response patterns reflected very closely the peptide sequence similarities (Supplementary Fig. 3).

**Identification of RV species-specific antibody increases.** Based on our previous observations that antibody increases specific for the N-terminal portion of VP1 can be detected in serum samples obtained from subjects after RV infection[36], the PreDicta chip was equipped with a VP1 peptide set which should allow detecting species-specific immune responses at high resolution (Fig. 1). Supplementary Figure 4 shows images of RV-specific antibody responses measured in sera from six representative children obtained at the time of the acute episode of wheeze and in sera at a follow-up visit 2–3 months later. In sera from the follow-up visit increased antibody responses to RV-A (sera #33, #108), RV-C (sera #119, #84), and RV-B (sera #66, #89) peptides could be detected (Supplementary Fig. 4).

We then compared the peptide-specific IgG antibody levels in sera obtained from the 120 children at the time of the acute wheeze and at the follow-up 2–3 months later (i.e., median 11 weeks later) (Table 1). Supplementary Figure 5 shows a color-coded map of the peptide-specific antibody responses for the acute phase and follow-up of each of the children demonstrating species-specific IgG increases. Next, we com-pared the peptide-specific increases in relative numbers for each child (Fig. 3a). According to these peptide-specific IgG increases, children could be identified who responded prefer-entially to RV-A-derived peptides (n = 41), RV-C-derived peptides (n = 33), RV-B-derived peptides (n = 23), and some with a mixed response pattern (n = 7) (Fig. 3a). For 16 children no increases of peptide-specific IgG responses were found (Fig. 3a, bottom). The same analysis was performed for increases of antibody responses against complete recombinant VP1 proteins from the three RV-A strains (i.e., RV89, 16, 2),

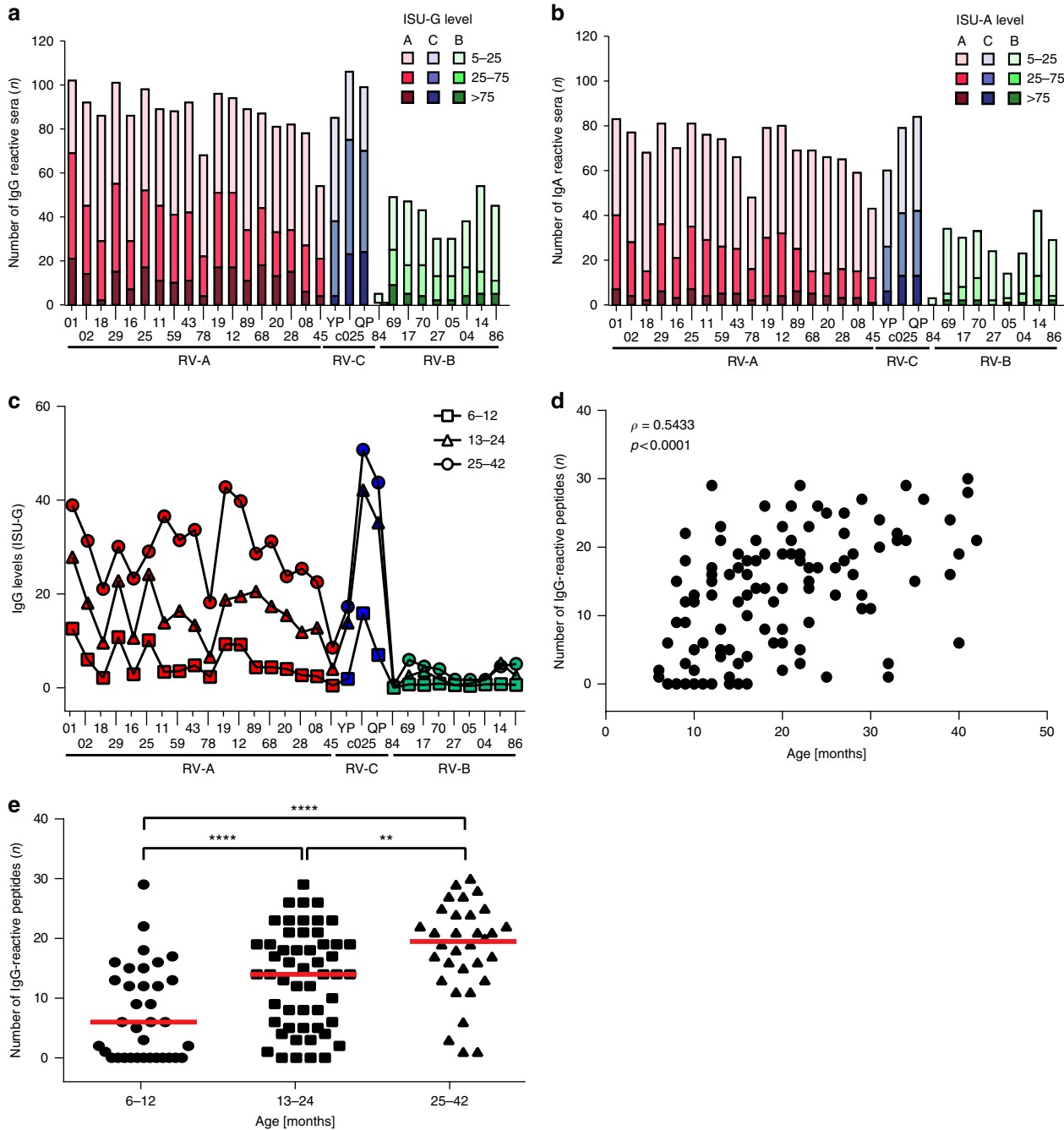

**Fig. 2** RV-specific antibody responses in sera from children with acute wheeze. Frequencies and levels of **a** IgG and **b** IgA responses (y-axes: n, number of reactive sera) to the N-terminal VP1 peptides from 30 RV strains (Supplementary Tables 2 and 4) (x-axes: red: RV-A species; green: RV-B species; blue: RV-C species). Antibody levels are color-coded and expressed as ISAC standardized units, ISU-G and ISU-A, respectively. **c** Median IgG levels (y-axis: ISU-G) to VP1 peptides (x-axis) in children grouped according to age (6–12 months: squares; 13–24 months: triangles; 25–42 months: circles). **d** Spearman's rank correlation between the number of IgG-reactive peptides (n, y-axis; median IgG >15 ISU) and age (x-axis: months). Correlation coefficient ($\rho$) and p-value are shown. **e** Comparison of the number of IgG-reactive VP1 peptides (n, y-axis; median IgG >15 ISU) in children according to age (x-axis). Horizontal lines indicate medians. Statistically significant differences between groups are indicated (**$p < 0.01$, ****$p < 0.0001$) (Mann–Whitney U-test)

the RV-C strain YP, and the RV-B strain 14 (Fig. 3a) but the results were less clear and negative for several children because only few strains were covered with the recombinant proteins. We have also included in Fig. 3a (right column) results from the PCR testing performed using VP4–VP2-specific primers in 108 of the 120 children[35], which showed that the nucleic acid-based detection of virus strains was negative for approximately 25% of children with increases of RV peptide-specific IgG levels which may indicate a higher

sensitivity of serology vs. PCR in these children. Moreover, PCR results did not correspond well with the specificities of the antibody responses. There were also 14 children without increases of RV peptide-specific antibody responses who had positive PCR results (Fig. 3a, bottom).

Figure 3b provides a summary of the absolute increases of peptide-specific IgG responses showing that RV-A and RV-C-specific antibody responses dominated over RV-B-specific increases.

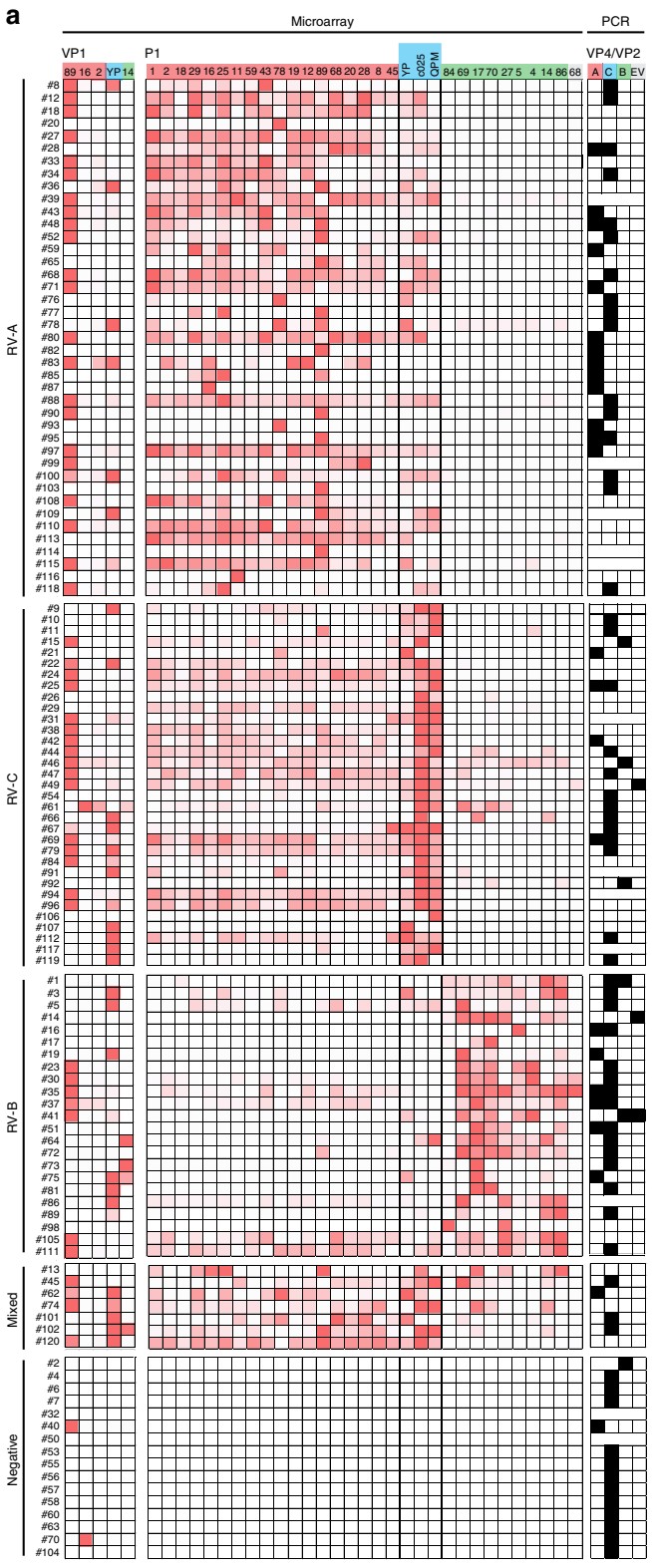

**Fig. 3** Increases of IgG antibodies to N-terminal VP1 peptides detect culprit RV species. **a** A map showing IgG antibody increases (gradient color scale ranging from minimal to maximal value for each of the subject: white to dark red) to recombinant VP1 proteins and N-terminal VP1 peptides from three RV species (top lines RV-A: red; RV-C: blue; RV-B: green) measured by the microarray and PCR data for the wheezing children (white squares: PCR negative; black squares: PCR positive; no squares: not tested due to lack of samples). Shown are children for which RV-A ($n = 41$), RV-C ($n = 33$), and RV-B ($n = 23$) were identified as culprit species according to chip analysis, as well as children with mixed ($n = 7$) and no responses ($n = 15$). **b** Absolute increases of IgG antibody levels between the acute and follow-up visit for the wheezing children (y-axes: $n$, number of reactive sera) to the VP1 peptides (x-axes: red: RV-A species; green: RV-B species; blue: RV-C species) are shown. Antibody increases are color-coded and expressed as ISAC standardized units (ISU-G)

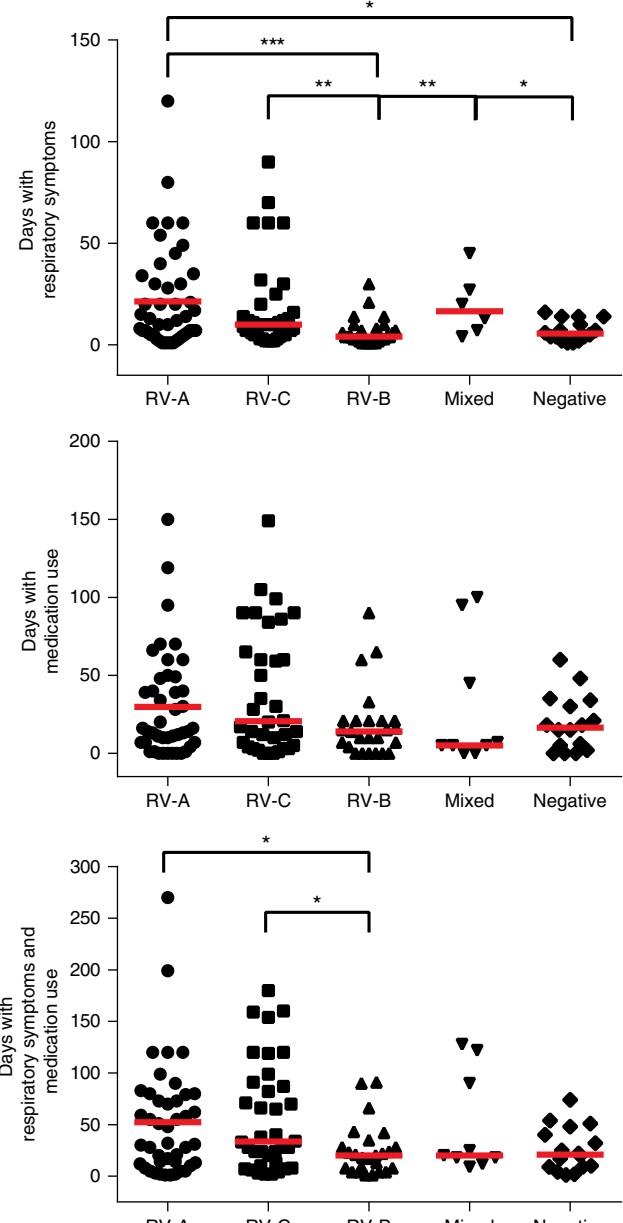

**Fig. 4** Days with respiratory symptoms and medication use in children without and with RV-A, RV-B, RV-C, or mixed peptide-specific IgG increases. Days with respiratory symptoms (top panel, *y*-axis), with medication use (middle panel, *y*-axis), and with symptoms and medication use (bottom panel, *y*-axis) are shown for children without (negative), mixed, RV-A, RV-C, or RV-B increases of IgG (*x*-axes). Horizontal lines show median values. Statistically significant differences between groups are indicated (***$p < 0.001$; **$p < 0.01$; *$p < 0.05$) (Mann–Whitney *U*-test)

**Antibody signatures associated with severity of wheeze.** Next, we investigated whether antibody responses to certain RV species were associated with the severity of RV-induced wheeze. For this purpose, we determined the number of days with respiratory symptoms and the number of days when medication was required in the period between the acute and follow-up visit. The number of days with respiratory symptoms but not with medication was significantly higher in subjects with an increase in RV-A > Mixed > RV-C > RV-B specific signal (Fig. 4, top and middle panels). We then analyzed the sum of days with symptoms and medication and found that RV-A and RV-C antibody increases were

associated with the highest number of days with symptoms and medication (Fig. 4, bottom panel). Children with RV-A- or RV-C-specific antibody increases had significantly more days with symptoms and medication than children with RV-B-specific antibody increases (Fig. 4, bottom panel).

## Discussion

Within the European Union-funded project PreDicta (https://cordis.europa.eu/project/rcn/96868_en.html) we developed the PreDicta chip which is based on 130 micro-arrayed RV-derived proteins and peptides selected to represent the three RV species (RV-A, RV-B, and RV-C). Using the PreDicta chip we could demonstrate in a cohort of 120 pre-school children with acute wheeze and a follow-up visit that the RV-specific antibody response (IgG > IgA) is directed against an N-terminal peptide of the major capsid protein VP1 which confirms earlier results obtained by the mapping of RV89-specific antibody responses[33]. Since the PreDicta chip was designed to contain a panel of 30 synthetic peptides which represented the most diverse RV strains of the three genetic RV species in terms of sequence identity and physicochemical properties we were able to perform a high-resolution mapping of RV species-specific antibody responses by serology. In a cohort of clinically well-described Swedish pre-school children from whom sera were available from an episode of acute wheeze requiring an emergency room visit and from a follow-up visit after convalescence, peptides from RV-A and RV-C species were most frequently recognized whereas RV-B species were much less commonly recognized. Interestingly, we found that older children (i.e., children older than 2 years of age) recognized peptides from more RV strains than younger children. This result would indicate that children encounter in their life different RV strains and thus may broaden their IgG reactivity profiles later in life but this needs to be confirmed in longitudinal studies with samples taken from the same children at different ages, as has recently been done in the analysis of the evolution of IgE reactivity profiles in allergic children in birth cohorts[37–40]. One of the important findings of our study was that we could demonstrate that IgG reactivity to peptides from certain RV strains increased in the children which may allow identifying the culprit RV species responsible for the acute wheeze by serology. Furthermore, it turned out that increases of IgG responses to RV-A and RV-C species were significantly associated with more severe illness as compared to IgG increases to RV-B. The PreDicta chip thus seems to be not only suitable for identifying the culprit RV species responsible for an exacerbation of respiratory illness by simple serology but also allows to determine those RV species giving rise to severe symptoms. Since we also had results from PCR-based testing of nasal swab samples from the same children we could compare the detection of strain-specific nucleic acid with antibody results. In fact, we found that all children without species-specific antibody increases were also negative in the PCR test. However, there was poor correlation between the PCR results from the nasal samples and the chip-based serological results. Furthermore, with the exception of four children (#3, #61, #65, #72) for which a positive PCR result has been obtained at the follow-up visit, all PCR results were negative at both the acute and follow-up visit in approximately 25% of the children for whom species-specific antibody increases could be clearly demonstrated. Several possibilities for the discrepancies may be considered. For example, it may be possible that the time interval after acute infection chosen for serology was not optimal. However, we have previously investigated the time interval required for the appearance of VP1-specific antibody increases in a controlled infection study[36] and it is therefore

very unlikely that the time interval used in our study was too long and responsible for discrepancies between PCR testing and serology. We also do not think that cross-reactivity among strains or the original antigenic sin[41] is responsible for the observed differences because we have taken care to include on the chip sequences from several different RV strains and the results in Fig. 3a clearly show that the serological results obtained with the chip allow a bona fide discrimination of RV-A, RV-B, and RV-C infections because we have used a large panel of RV peptides on the chip. Finally, we found that older children recognized more RV peptides from different strains than younger children (Fig. 2c–e), which indicates that the children develop antibodies against new viruses and thus the concept of the original antigenic sin does not seem to apply here. One limitation of our study is, however, that we cannot exclude that infections with additional RV strains have occurred in the time window between the first and second blood sampling and thus are responsible for the discrepancy between PCR results and serology. Nevertheless we think that the discrepancy between PCR results and serology is rather due to the fact that not every virus detected by PCR causes an infection with a consecutive immune response. In fact, studies performed in young children report that up to 35% of asymptomatic subjects have positive PCR results[30,42]. One more likely possibility for the poor correlation of PCR results and antibody results could be that we used a PCR strategy based on primers specific for VP4- and VP2-encoding regions of the viral genome, which may be less specific than PCR strategies based on the amplification and sequencing of the VP1-encoding region or of the complete RV genome[12,28,43,44]. In general, nucleic acid-based strategies for virus detection only demonstrate the presence of virus-specific nucleic acid but provide no evidence that an infection has taken place which gave rise to a specific immune response. We therefore think that the PreDicta chip and future versions of it containing an even larger repertoire of N-terminal VP1 peptides from more RV strains will be a complementary tool in addition to PCR testing and eventually turn out to be even superior. The antibody test is actually fast and economical: It takes only few hours and requires only microliter amounts of serum. Furthermore, serological analysis is robust and can be easily performed without need for PCR cyclers and subsequent sequencing. However, more prospective studies will be needed to investigate the diagnostic sensitivity and specificity of the RV chip. The PreDicta chip may be extremely useful to determine RV species-specific antibody responses in serum samples from existing cohorts world-wide to define the most common and relevant RV species involved in respiratory illness. Chip-based measurements will allow exploring in prospective studies the role of RV infections in a variety of respiratory disease exacerbations. For example, it will be possible to discriminate whether asthma exacerbations have been triggered by an RV infection or by allergen exposure because it has been shown that both factors (i.e., RV infections as well as allergen exposure) induce increases of specific antibody responses when serum samples collected at the acute visit and during a follow-up visit are compared[35,36,45,46]. The identification of the culprit factors triggering asthma attacks becomes increasingly important in respiratory medicine due to availability of selective treatments of allergic asthma such as anti-IgE antibodies and a variety of other biologics such as anti-cytokine antibodies targeting different forms of asthma[47,48]. Furthermore, it will be possible to use the PreDicta chip to study by serological analysis the possible contribution of RV infections in exacerbations of other respiratory diseases such as COPD and ACO. Further studies are also necessary to perform a multiple monitoring of the presence of RV strains and other

respiratory viruses by PCR and the immune reaction by serology in close intervals and for extended periods after exacerbation.

The reliable determination of the most common RV species involved in triggering severe respiratory illness will ultimately provide a rational basis for the development of RV vaccines and RV species-targeting therapeutic approaches[20–26].

In conclusion, we developed and evaluated a high-resolution antibody assay based on micro-arrayed peptides and recombinant antigens from the most common RV strains to identify antibody signatures discriminating RV infections at the levels of different RV species and allowed to point towards the culprit species responsible for the triggering of acute pre-school wheezing. The PreDicta chip has the potential to be useful for a serological global mapping of RV infections, the identification of RV species involved in triggering different forms of severe respiratory illness, and for paving the road for RV-specific therapeutic and prophylactic treatment strategies, such as vaccines.

## Methods

**Selection and production of N-terminal VP1 peptides**. VP1 amino acid sequences of RV strains representing the three RV species (RV-A: $n = 76$; RV-B: $n = 25$; RV-C: $n = 6$) were retrieved from the NCBI database (https://www.ncbi.nlm.nih.gov/) (Supplementary Table 1). Multiple sequence alignments of the VP1 N-terminal peptides (aa 1–40) were performed using ClustalW2 software available at the EMBL-EBI website (http://www.ebi.ac.uk/tools/clustalw2) to determine amino acid sequence identities among the peptides. Peptide sequences showing sequence identities greater than 87.5% (i.e. differences of ≤5 aa) were grouped together into clusters (Fig. 1). Sequences among the clusters (Supplementary Tables 1 and 2, A1–A17, B1–B9, C1–C3) were re-aligned using GeneDoc software (http://iubio.bio.indiana.edu/soft/molbio/ibmpc/genedoc-readme.html) and each amino acid mismatch was analyzed regarding physicochemical properties of the amino acids. This procedure led to re-clustering of the peptides (Supplementary Table 2, AI–XVIII, BI–BIX; CI–CIII). From each re-clustered group one representative RV strain peptide was selected for printing onto the chip (Fig. 1b, Supplementary Tables 2 and 3: RV-A: $n = 18$; RV-B: $n = 9$; RV-C: $n = 3$). An enterovirus-derived peptide was also included (Fig. 1). For the set of peptides to be printed a multiple sequence alignment was performed by ClustalW2 and a phylogenetic tree was constructed by the Neighbor-Joining (N-J) method using MEGA 6 software (www.megasoftware.net) (Fig. 1c). The evolutionary distances between sequences were computed using the Kimura 2-parameter model with bootstrap values calculated from 1000 replicates. Additional 36 peptides spanning the RV89VP1 protein (Fig. 1, Supplementary Table 4) were selected to detect antibodies towards VP1 epitopes other than the N-terminal portion.

The peptides as well as the non-structural 3B protein from strain 89 (VPg: GPYSGEPKPKSRAPERRVVTQ) were produced by solid-phase synthesis with the 9-fluorenyl-methoxy carbonyl (Fmoc)-method (CEM-Liberty, Matthews, NC, USA and Applied Biosystems, Carlsbad, CA, USA) on PEG-PS preloaded resins (Applied Biosystems). After synthesis, peptides were washed with dichloromethane, cleaved from the resins using 19 ml trifluoroacetic acid (TFA), 0.5 ml silane, and 0.5 ml H$_2$O and precipitated into pre-chilled tert-butylmethylether. Peptides were purified by reversed-phase high-performance liquid chromatography in a 10–70% acetonitrile gradient using a Jupiter 4 μm Proteo 90 Å, LC column (Phenomenex, Torrance, CA, USA) and an UltiMate 3000 Pump (Dionex, Sunnyvale, CA, USA) to a purity >90%. Their identities and molecular weights were verified by mass spectrometry (Microflex MALDI-TOF, Bruker, Billerica, MA, USA)[49].

For the unsupervised analysis of antibody responses to RV peptides and proteins, unsupervised K-means clustering ($K = 4$) was used to define clusters of peptides with similar antibody response measurements. The K number of clusters was pre-determined in order to match the number of the peptide homology groups. The peptides' amino acid sequences were aligned with a Gap open cost of 10.0 and a Gap extension cost of 1.0. Based on the alignment, a homology distance cladogram was built using the Neighbor-Joining algorithm and 1000 bootstrap replicates. The peptides were then color-coded based on the antibody response cluster that they belonged to. Data were processed using the CLC Genomics Workbench (CLC, CLCbio, Qiagen, Hilden, Germany). The heat map representing RV-specific antibody responses was generated by Qlucore Omics Explorer (Qlucore, Lund, Sweden).

**Expression and purification of recombinant RV proteins**. Recombinant his-tagged structural (VP1–4) proteins from five representative RV strains (RV-A2, -A16, -A89, -B14, and -CYP) and MBP fusion proteins containing fragments thereof (VP1–3) as well as non-structural (2A, 2C, 3A, 3C, and 3D) proteins from RV strain 89 were expressed in *Escherichia coli* as previously described[33,36]. DNA sequences coding for the complete genes or fragments thereof (accession numbers

are shown in Supplementary Table 5) were codon optimized for bacterial expression, synthesized with the addition of the 3′ sequence coding for a C-terminal hexa-histidine tag and cloned into the *Nde*I and *Eco*RI sites of plasmid pET27b (Genscript, Piscataway, NJ, USA). Transformed *E. coli* BL21 (DE3) cells (Agilent Technologies, Santa Clara, CA, USA) were induced with 1 mM isopropyl-β-thiogalactopyranoside (IPTG) and cells were harvested at time-points of maximal expression. Recombinant proteins were purified by Nickel-affinity chromatography under denaturing conditions as previously described (Qiagen, Hilden, Germany). Refolding of recombinant proteins was achieved by a stepwise dialysis against 10 mM $NaH_2PO_4$ for structural and non-structural proteins and 20 mM Tris-HCl, 200 mM NaCl, 1 mM EDTA for MBP fusion proteins, respectively. The purity of recombinant proteins was verified by SDS-PAGE followed by Coomassie Brilliant Blue staining and the identity by immunoblotting using a monoclonal mouse anti-His-tag antibody 1:1000 diluted (Cat: DIA-900, Dianova, Hamburg, Germany). Bound antibodies were detected with 1:1000 diluted alkaline phosphatase-coupled rat anti-mouse IgG antibodies (Cat: 557272; BD Biosciences, Erembodegem, Belgium). Protein concentrations were determined using BCA Protein Assay Kit (Thermo Fisher Scientific, Rockford, IL, USA). The secondary structure of the proteins was measured by circular dichroism spectroscopy on a Jasko J-810 spectropolarimeter (Japan Spectroscopic, Tokyo, Japan) at a protein concentration of 0.1 mg/ml in 10 mM $NaH_2PO_4$.

**Detection antibodies and printing of microarrays**. Anti-huIgG (Cat: 309-005-008; Jackson ImmunoResearch Laboratories, West Grove, PA, USA) and anti-huIgA (Cat.: 555885; Becton Dickinson, Franklin Lakes, NJ) were labeled with DyLight 650 (Pierce, Thermo Fisher Scientific, Rockford, IL, USA). Customized printing of RV microarrays was done by Phadia-ThermoFisher using ImmunoCAP ISAC (Immuno Solid-phase Allergen Chip) technology[50,51]. Spotting was performed by slow pin mode printing using the Aushon 2410 Printer (Aushon, Billerica, MA, USA). Stock solutions of peptides (5 mg/ml) were diluted 1:4 in a phosphate buffer, pH 8.4 and then used for spotting. Antigens were spotted in triplicates on a glass surface coated with an amino-reactive organic polymer, each spot containing 50–200 fg of microarray component, corresponding to 1–5 attomol. Allergens used for the calibration and other control proteins spotted on the microarray are listed in Supplementary Table 6.

**Cohort of pre-school children with acute wheeze**. Serum samples examined in this study were from a cohort of 120 pre-school children who had been admitted to the Paediatric Emergency Ward as a result of acute wheeze, at Astrid Lindgren Children's Hospital, Stockholm, Sweden (Table 1). This cohort and the genotyping of RV strains in the nasopharyngeal swab samples of 108 of the 120 children by nested PCR and sequencing have been previously described[35]. A molecular diagnostic platform for the rapid detection of 15 respiratory strains was used in 118 of the 120 children. The following respiratory viruses were found: Adenovirus: 7 children; Bocavirus: 8 children; Coronavirus: 6 children; Influenza A/B: 1 child; Metapneumovirus: 3 children; Parainfluenzavirus: 4 children: RSV: 22 children[52]. For 108 of the 120 children nasopharyngeal swabs had been available for the RV PCR targeting VP4/2 sequences. Written informed consent was obtained from the parents or by the legal guardians and the study was approved by the Regional Ethics Committee of Karolinska Institutet, Stockholm, Sweden. Peripheral blood samples had been obtained within 24 h of presentation in the emergency unit and sera were stored at −80 °C. In addition to blood samples, nasopharyngeal swab samples were obtained at the acute visit and again at the follow-up visit by the research nurse and stored in the biobank at the Department of Clinical Microbiology, Karolinska University Hospital[35]. Follow-up samples were obtained between 6 and 30 weeks after the initial recruitment (median 11 weeks) at a scheduled visit after recovery. Although this study was not planned as a prospective study for the assessment of increases of RV-specific antibodies, the time interval of 6–30 weeks was suited for this purpose because we found in an earlier study that increase of RV-specific IgG responses emerge 42 days after experimental inoculation[36]. At the follow-up visit, the guardians also filled out a standardized questionnaire concerning the number of days the child had suffered from respiratory symptoms at home (i.e., 'cough and/or wheeze'), use of medication (i.e., β2-agonists, inhaled corticosteroids, leukotriene receptor antagonist), as well as about any emergency visits between the acute and follow-up visits. The chip analysis of the anonymized sera was performed with the approval of the ethics committee of Medical University of Vienna (EK1721/2014)[37,50].

**Microarray-based determination of antibody profiles**. Microarrays were washed in a washing buffer (Phadia-Thermo Fisher) for 5 min by stirring. After drying by centrifugation (1 min, 1000 *g*, RT), 35 µl of serum samples were applied on each microarray and the slides were incubated for 2 h at gentle rocking (RT). For the detection of RV-specific IgG and IgA antibodies, serum samples were diluted 1:300 and 1:20 in a sample dilution buffer (Phadia-Thermo Fisher), respectively. Microarrays were then rinsed with washing buffer and washed for 5 min as described above. After centrifugation, 30 µl of fluorescence-labeled antibodies (1 µg/ml) was added and the slides were incubated 30 min at gentle rocking (in dark, RT). After further rinsing, washing and drying, microarrays were scanned using a confocal laser scanner (LuxScan-10K microarray scanner, Capital-Bio, Beijing,

People's Republic of China) and the image analysis was evaluated by Microarray Image Analyzer v3.1.2 software (Phadia-Thermo Fisher)[50]. For calibration and determination of background signals, a calibrator serum (i.e., a pool of allergic patients sera, diluted 1:10) and sample diluent were included in each analysis run.

**Calibration and variability of the microarray**. The Phadia Microarray Image Analysis software was used to process images, to calculate the mean fluorescence intensities (FI) of triplicate analyses and to calibrate the results. A calibration curve was generated by relating fluorescence intensities obtained by scanning the Pre-Dicta microarray with allergen-specific antibody levels measured by ImmunoCAP. Results were reported in ISAC standardized units (ISU)[50,52]. Background reactivities of fluorescence-labeled α-huIgG antibodies towards all microarray components were determined by testing six replicates of a sample diluent alone. For the characterization of the assay variability, one calibrator serum and three normal sera were profiled at 1:100, 1:200, 1:500, and 1:1000 in order to find a dilution at which the broadest spectrum of reactivity levels were covered. To assess intra-assay (i.e. within experiment) variability, six replicates of four serum samples were used to measure IgG reactivities towards all microarray components on the same day. To assess inter-assay (i.e. between experiments) variability, four serum samples were evaluated in experiments conducted on five consecutive days. The evaluation of four different samples allowed determining whether the inter-assay variability was sample-dependent. For each microarray component, the mean ISU-G, standard deviation (SD), coefficient of variation (CV % = SD/mean ISU-G) across the six replicates and five different experiments, respectively, were calculated for each sample. Microarray components were classified according to the ISU-G level (>50, 25–50, 1–25) and averages across all components within these groups on the array were also calculated for each of these quantities.

**Data analysis**. Initial data processing was performed with Microsoft Excel. Frequencies (i.e., the number of reactive sera) and intensities (i.e., ISU-G levels) of peptide-specific IgG and IgA antibody levels were calculated using IBM SPSS Statistics (version 24; IBM Corp., Armonk, NY, USA). Median values of peptide-specific IgG levels (ISU-G) were calculated using GraphPad Prism 6 (La Jolla, CA, USA). The cut-off for a positive IgG reactivity was set at 1 ISU-G. The numbers of IgG-reactive peptides in Fig. 2d, e were calculated for specific IgG levels >15 ISU-G. Increases of RV-specific antibody responses were calculated as differences between ISU-G values measured at the follow-up (F) and the acute (A) visits (ΔISU-G = ISU-$G_F$ − ISU-$G_A$) followed by subtraction of the double coefficient of variation (2 × CV%) calculated from the baseline values (ISU-$G_A$) for each of the antigens. For IgG levels of 1–5 ISU-G and >5 ISU-G, 25% and 12.5% were determined as CV% for intra-assay variability and subtracted from the data, respectively. RV-specific IgM antibodies were not measured because we found in an earlier study that no increases of RV-specific IgM antibody levels were detectable on days 4, 7, and 42 after experimental RV inoculation[36].

**Statistical analysis**. GraphPad Prism 6 (La Jolla, CA, USA) was used to evaluate all statistical parameters. Correlation between the number of reactive peptides and age was evaluated by calculating the Spearman's rank correlation coefficient ($\rho$). Differences between groups in number of reactive peptides and in number of days the children spent with respiratory symptoms, medication use, or with both were assessed by Mann–Whitney *U*-test (two-tailed). Values of $p < 0.05$ were considered statistically significant.

**Data availability**. Primary data that support the findings of this study are available from the corresponding author on request.

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

## Acknowledgements

This study was funded by PreDicta, a FP7-funded EU project (No. 260895), by the FWF-funded projects F4605 and P29398 of the Austrian Science Fund, by research grants from Biomay AG and Viravaxx, Vienna, by the Swedish Research Council, the Swedish Heart-Lung Foundation, Stockholm County Council (ALF project), the Swedish Asthma and Allergy Association´s Research Foundation, the King Gustaf V´s 80-year Foundation, the Centre for Allergy Research at Karolinska Institutet and the Swedish Cancer and Allergy Foundation.

## Author contributions

Conception and design: K.N., S.M., N.G.P., and R.V.; analysis and interpretation: K.N., K.S.-H., S.M., C.R.C., K.N.-W., P.C.V., D.G., C.L., D.E., T.S., .C.H., C.S., M.v.H., G.H., N.G.P., and R.V. Drafting and or reading the manuscript for important intellectual content: K.N., K.S.-H., S.M., C.R.C., K.N.W., P.C.V., D.G., C.L., D.E., T.S., .C.H., C.S., M.v.H., G.H., N.G.P., and R.V.

## Additional information

**Competing interests:** C.L. and M.v.H. report personal fees from Thermo Fisher Scientific, outside the submitted work. M.v.H. serves as a consultant for Biomay AG, Vienna, Austria and Hycor Biomedical LLC, CA, US. N.G.P. reports personal fees from

Novartis, Faes Farma, Biomay AG, Vienna, Austria, HAL, Nutricia Research, Menarini, MEDA, Abbvie, MSD, Omega Pharma, Danone, grants from Menarini, outside the submitted work. R.V. reports grants from European Union, grants and personal fees from Biomay AG, Vienna, Austria, grants and personal fees from Viravaxx, Vienna, Austria, during the conduct of the study; grants from Austrian Science Fund (FWF), grants and personal fees from Biomay AG, Vienna, Austria, grants and personal fees from Viravaxx AG, Vienna, Austria, outside the submitted work. In addition, R.V. and K.N. are co-inventors in a patent application (PCT/AT2010/000416) regarding the rhinovirus diagnosis reported in this paper. The remaining authors declare no competing interests.

