## [Peer Review File · Nature Communications]

Reviewers' comments:

Reviewer #1 (Remarks to the Author):

This study reports a chip for the serological diagnosis of rhinovirus-induced respiratory infection. The topic is very important. A reliable and easy serological test for specific IgG response is highly needed to verify the real role of rhinovirus infection e.g. in asthma, pneumonia in children and adults as well as in severe infections of COPD patients. Problems with the etiology of pneumonia should be mentioned in the paper.

I find some major problems in this study.

The study is clearly not prospectively planned for serological measurements. The interval of paired serum samples in serological studies is usually and should be 2-4 weeks and in this study the mean interval was 2-3 months. This certainly permits occurrence of other respiratory virus infections during the follow-up. I miss more data on the kinetics of IgG responses. I also miss IgM responses, only a comment.

In this study, PCR test for rhinovirus was performed in 108 out of 120 children. Originally, a more sensitive diagnostic PCR targeting the 5'NCR should have been used, which may have given a higher recovery. For validation and specificity of the serological test, I miss the detection results of other respiratory viruses causing wheezing illness in children e.g. RSV, enteroviruses, human bocavirus, adenovirus, human metapneumovirus.

I am confused of the comment on page 9: "the nucleic acid based detection of virus strains was negative for appr. 25% of children with increases of RV-peptide-specific IgG levels. Moreover, PCR results did not correspond well with the specificities of the antibody responses." PCR results for rhinovirus infection are currently considered as gold standard but the sensitivities of different assays may vary. So, comparison against PCR results would be meaningful. Although the sensitivity of VP2/4 PCR may not be as good as that of 5'NCR, VP2/4 sequences are considered as specific for RV species and known type as those of VP1 (ref. 13). Therefore, the VP2/4 PCR result of the acute phase specimen should be considered the correct result and the reason for disagreement between that and the serology should be sought from the latter. The discrepancies may be due to long interval between serum samples and the possibility of subsequent infections with different RV strains during that time. Cross reactivity is also likely and it would be surprising if the phenomenon of the original antigenic sin would not play any role in the serology of RV infections.

The authors seem over enthusiastic about the diagnostic power of species specific serology. It is

much needed for research of RV infections, e.g. in clinical studies of potential drugs and vaccines, but for the diagnosis in the acute phase of infection it is probably useless, and delay in collection of paired samples makes it impractical for routine clinical purposes.

Reviewer #2 (Remarks to the Author):

This paper describes construction and application of a protein microarray consisting of a diversity of Rhinovirus recombinant proteins and peptides representing different Rhinovirus subtypes. The arrays can be probed with serum from individual exposed to the virus and relative Ab levels between clinically distinct groups accurately and reproducibly compared.

Although the concept of using protein microarrays for this type of seroprevalence study is not novel, the particular Rhino virus protein/peptide array constructed here is novel. This paper highlights the practical, serum sparing, quantitative and reproducible features of protein microarrays to quantify Ab levels for this type of study - more practical than developing individual ELISA assays for dozens of individual antigen of interest.

The work shows convincingly that Rhinovirus subtypes differ in their seroprevalence, that seroprevalence against Rhinovirus antigens increases with age, and that Ab levels increase after a Rhinovirus induced wheezing episode. Age dependent increases in the antibody responses were noted with reactivity against species RV-C > RV-A > RV-B.

More papers of this kind are needed to show the practicality of protein microarrays for determining exposure and seroprevalence to infectious agents.

The www.predicta.eu link cited in the introduction, the source of funding for this work, isn't a very helpful source of background information about this project.

The manuscript lacks details about methods. What type of printer was used. What substrate were the peptides printed on. What is the concentration of peptide and what buffer.

I feel that the results in the manuscript can't be understood without liberal reference to information and figures in the supplement. Some of the most informative results are in the supplementary figures.

Supplementary Table 7 seems to be showing assay reproducibility data but it is not entirely clear

just from the table what has been done.

How do you know that IgG is lower than IgA? Only that the signal intensities (ISU) are lower for IgG than IgG.

It will be helpful to provide more explanation of ISAC and ISU to save the reader time looking up other papers, some not so accessible.

Supplementary Figure 3 is referred to and highlighted in the results section, but the important figure need to be looked up in the supplement.

I found it inconvenient to have to go back and forth between the narrative and the supplement to follow through the results section.

Legend to Figure 4 needs more detail.

Reviewer #3 (Remarks to the Author):

The authors have set out to generate a microarray based serologic test that can discriminate between RV-A,B,C RV species, as culprit species during acute asthma episodes in children. This article builds on previous work by a number of the authors that has shown that a strain specific increase in IgG1 anti-VP-1 antibodies peaks by day 42 post inoculation/infection of adult volunteers (healthy and with mild and moderate asthma). They and others have also shown an association with increased IgG1 antibodies to be seen in those with asthma and to correlate to severity of symptoms.

In this study they demonstrate in 120 children with acute asthma a rise in IgG and IgA antibodies that identifies RV species and again show those with the greatest increase in antibodies are associated with more severe clinical disease. There is also an age specific response, that increases with age that is not surprising and presumably reflects increasing maturity of the immune response or a cumulative exposure to RV species that increases with time.

The authors propose that this serological test can discriminate the RV species responsible for acute asthma in children and has the potential to map RV signatures in larger populations and in different respiratory diseases.

I broadly support their conclusions, though have a number of points that require discussion.

There were 32/108 children who were PCR negative for RV during their acute presentation, despite seeing a rise in RV antibodies. This is a large proportion. Given the long lag in antibody rise (42 days), prior exposure to other RV species may be responsible for the rise in antibodies, species that lead to subclinical infection. Alternatively, could there have been cross reactivity with other viruses, especially the related enterovirus family (especially EV-68), that can cause similar presentations to RV? More detail about this cohort would be required. Clinical severity, age, presence or absence of virus symptoms (granted this is difficult in this age group), time from symptoms onset to presentation with acute asthma. Indeed there are limitations to the use of viral swabs and PCR, poor collection techniques, degradation of specimens with storage or delay in

transport, reduced sensitivity in detection where virus transport media is used. It would argue for the repeatability and better internal consistency of a serological test.

While promising the applicability of these results to other age groups and disease states would require validation in these groups and is perhaps somewhat overstated in the manuscript. Despite the preliminary work done in adult volunteers, immune responses to viruses will be quite different in older age groups with both asthma and COPD, virus carriage and shedding is usually less and presentation with an exacerbation may be more delayed.

The manuscript is well written.

The statistics used appear appropriate.

The figures were of high quality.

Minor points

1. Page 3, line 16, I would regard CDHR3 as at least one of the probable receptors for RV-C, not possible.
2. Page 4, line 84, ACOS, GINA now recommends the term Asthma COPD overlap, without syndrome and would recommend using this term throughout the manuscript.

Questions for the authors

1. In those who were PCR negative for RV at presentation, were the differences in age, sex, symptoms of a virus infection, medication use or time from symptom onset to presentation with acute asthma?
2. Where other viruses investigated for at the time of presentation? In particular enteroviruses where a cross reactivity may occur?

Peter Wark

Point by point reply

Reviewer #1 (Remarks to the Author):

This study reports a chip for the serological diagnosis of rhinovirus-induced respiratory infection. The topic is very important. A reliable and easy serological test for specific IgG response is highly needed to verify the real role of rhinovirus infection e.g. in asthma, pneumonia in children and adults as well as in severe infections of COPD patients. Problems with the etiology of pneumonia should be mentioned in the paper.

REPLY: We thank the reviewer for these positive comments. According to the reviewer's suggestion, we mentioned the problems with the etiology of pneumonia in the revised manuscript (see lines 52-53 and 87-88).

I find some major problems in this study.

The study is clearly not prospectively planned for serological measurements. The interval of paired serum samples in serological studies is usually and should be 2-4 weeks and in this study the mean interval was 2-3 months. This certainly permits occurrence of other respiratory virus infections during the follow-up. I miss more data on the kinetics of IgG responses. I also miss IgM responses, only a comment.

REPLY: It is true that the study was not prospectively planned for serological measurements but we have previously performed a study which has allowed us to analyze the kinetics of RV-specific IgG and IgM responses in 28 asthmatic patients and 11 healthy individuals who had been experimentally infected with RV16 (Rhinovirus-induced VP1-specific Antibodies are Group-specific and Associated With Severity of Respiratory Symptoms. Niespodziana K, Cabauatan CR, Jackson DJ, Gallerano D, Trujillo-Torralbo B, Del Rosario A, Mallia P, Valenta R, Johnston SL. EBioMedicine. 2014 Nov 18;2(1):64-70). VP1-specific antibody levels were measured in serum samples obtained at days 0, 4, 7 and 42 but no relevant changes of VP1-specific reactivities were noted at days 4 and 7 and beginning increases of IgG antibody levels were only found in blood samples obtained 42 days (i.e., 6 weeks) after inoculation suggesting that the increases of RV-specific IgG antibodies develop much later than 2-4 weeks after infection. It therefore seems that the mean interval of 2-3 months investigated in our study was quite appropriate for detecting RV-specific IgG increases. We have added this information to the revised supplement (see lines 120-124).

Furthermore, we have performed a PCR analysis only at the follow-up visit, which was negative for all except for four children, but not repeated PCR tests in the follow up period to search for potential additional infections with other RV strains in this period. However, if an infection with other RV strains would have occurred in the period until the follow-up visit, one would expect according to the kinetics of IgG responses observed in our earlier study, that the induction of IgG antibodies against an additional RV strain would be much lower than to that strain which had caused the acute exacerbation at the first visit. Nevertheless, we mentioned in the revised discussion that it would be useful to conduct a prospective study in which we measure the RV-specific IgG antibodies in closer intervals and for a longer period after an acute exacerbation and perform also repeated PCR assessments (see lines 317-319). We have not performed RV-specific IgM measurements because the analysis of VP1-specific IgM responses in the previous inoculation study showed that there were no increases of VP1-specific IgM antibodies on days 4, 7 and 42 after

experimental infection (see Figure below for the reviewer's convenience). This information was added to the revised supplement section (see lines 181-183).

Reviewer's Figure 1:

Reviewer's Figure 1. VP1-specific IgM responses in subjects inoculated with RV16. Shown are IgM antibody responses to VP1 (y-axis: optical density values) measured in serum samples obtained on days 0, 4, 7 and 42 after infection (x-axis) in the three subject groups (healthy individuals, mild and moderate asthmatics).

In this study, PCR test for rhinovirus was performed in 108 out of 120 children. Originally, a more sensitive diagnostic PCR targeting the 5 NCR should have been used, which may have given a higher recovery. For validation and specificity of the serological test, I miss the detection results of other respiratory viruses causing wheezing illness in children e.g. RSV, enteroviruses, human bocavirus, adenovirus, human metapneumovirus.

REPLY: We thank the reviewer for this comment. We decided to target VP2/4 sequences because they are considered as specific for RV species as also pointed out later by the reviewer which was more important than the sensitivity if one wishes to compare PCR testing with serological testing. Using the nested PCR method targeting VP2/4 sequences, we were able to detect RV in 82 out of 108 samples (recovery rate of 76%). Testing for other respiratory viruses which might have caused wheezing illness in these children had already been done in a previous study of Stenberg-Hammar *et al.* (Subnormal levels of vitamin D are associated with acute wheeze in young children. Stenberg Hammar K, Hedlin G, Konradsen JR, Nordlund B, Kull I, Giske CG, Pedroletti C, Söderhäll C, Melén E. *Acta Paediatr.* 2014 Aug; 103(8):856-61). In this study, a molecular diagnostic platform for the rapid detection of 15 respiratory strains was used in 118 of the 120 children. The following respiratory viruses were found: Adenovirus: 7 children; Bocavirus: 8 children; Coronavirus: 6 children; Influenza A/B: 1 child; Metapneumovirus: 3 children; Parainfluenzavirus: 4 children; RSV: 22 children. For 108 of the 120 children nasopharyngeal swabs had been available for the RV PCR targeting VP2/4 sequences and positive results for RV were obtained for 82 out of the 108 tested children. Thus RV was by far the most common respiratory virus detected in the children. This was mentioned in the revised supplement section (see lines 105-110).

I am confused of the comment on page 9: the nucleic acid based detection of virus strains was negative for appr. 25% of children with increases of RV-peptide-specific IgG levels.

REPLY: With the comment we wanted to explain that the serological chip test was more sensitive than the PCR test because 25% of children with increases of RV-peptide-specific IgG had a negative PCR test result. We have revised the statement to make it more understandable (see lines 210 – 211).

Moreover, PCR results did not correspond well with the specificities of the antibody responses. PCR results for rhinovirus infection are currently considered as gold standard but the sensitivities of different assays may vary. So, comparison against PCR results would be meaningful. Although the sensitivity of VP2/4 PCR may not be as good as that of 5 NCR, VP2/4 sequences are considered as specific for RV species and known type as those of VP1 (ref. 13). Therefore, the VP2/4 PCR result of the acute phase specimen should be considered the correct result and the reason for disagreement between that and the serology should be sought from the latter. The discrepancies may be due to long interval between serum samples and the possibility of subsequent infections with different RV strains during that time.

Cross reactivity is also likely and it would be surprising if the phenomenon of the original antigenic sin would not play any role in the serology of RV infections. The authors seem over enthusiastic about the diagnostic power of species specific serology. It is much needed for research of RV infections, e.g. in clinical studies of potential drugs and vaccines, but for the diagnosis in the acute phase of infection it is probably useless, and delay in collection of paired samples makes it impractical for routine clinical purposes.

REPLY: We thank the reviewer for the thoughtful comments. We agree that the VP2/4 PCR results are the most correct results if one performs PCR testing. This is exactly the reason why we selected the VP2/4 PCR strategy for comparison with serology. As explained in the reply above, we have previously investigated the time interval required for the appearance of VP1-specific antibody increases in a controlled infection study and it is therefore very unlikely that the time interval used in our study was too long and responsible for discrepancies between PCR testing and serology. We also do not think that cross-reactivity among strains or the original antigenic sin is responsible for the observed differences because we have taken care to include on the chip sequences from several different RV strains and the results in Figure 3a clearly show that the serological results obtained with the chip allow a *bona fide* discrimination of RV-A, RV-B and RV-C infections. Finally, we found that older children recognized more RV peptides from different strains than younger children (Figures 2c-e), which indicates that the children develop antibodies against new viruses and thus the concept of the original antigenic sin does not seem to apply here. We have also analyzed sera from longitudinal birth cohorts which show that children broaden their antibody response when they get older and have contact with new RV strains. We therefore think that the discrepancy between PCR results and serology is rather due to the fact that not every virus detected by PCR causes an infection with a consecutive immune response. In fact, studies performed in young children report that up to 35% of asymptomatic subjects have positive PCR results (Serial viral infections in infants with recurrent respiratory illnesses. Jartti T, Lee WM, Pappas T, Evans M, Lemanske RF Jr, Gern JE. Eur Respir J. 2008 Aug;32(2):314-2). Furthermore, PCR testing by nature is prone to errors because it involves a transcription and multiple amplification steps.

Importantly, there seems to be a big misunderstanding. We have never claimed to use the antibody-based test for the detection of acute infections. On the opposite, we propose to use the RV-specific antibody tests for a retrospective discrimination of different causes for respiratory exacerbations, to investigate the role of RV infections regarding the induction of exacerbations of

other chronic respiratory diseases such as COPD and ACO, to study the role of RV-induced exacerbations of respiratory diseases in different age groups (children, adults and elderly persons).

We have revised the discussion to include the interesting thoughts of the reviewer (see lines 269-288) and mentioned that more prospective studies will be needed to investigate the diagnostic power of the RV chip described in our study (see lines 301-302).

Reviewer #2 (Remarks to the Author):

This paper describes construction and application of a protein microarray consisting of a diversity of Rhinovirus recombinant proteins and peptides representing different Rhinovirus subtypes. The arrays can be probed with serum from individual exposed to the virus and relative Ab levels between clinically distinct groups accurately and reproducibly compared.

Although the concept of using protein microarrays for this type of seroprevalance study is not novel, the particular Rhino virus protein/peptide array constructed here is novel. This paper highlights the practical, serum sparing, quantitative and reproducible features of protein microarrays to quantify Ab levels for this type of study - more practical than developing individual ELISA assays for dozens of individual antigen of interest.

The work shows convincingly that Rhinovirus subtypes differ in their seroprevalance, that seroprevalance against Rhinovirus antigens increases with age, and that Ab levels increase after a Rhinovirus induced wheezing episode. Age dependent increases in the antibody responses were noted with reactivity against species RV-C > RV-A > RV-B.

More papers of this kind are needed to show the practicality of protein microarrays for determining exposure and seroprevalance to infectious agents.

REPLY: We thank the reviewer for these very positive comments and agree that more studies of this kind are needed to show the usefulness of protein microarrays for the detection of infectious agents in serum samples. In fact, such studies can now be performed with the micro-array described in our study and we think that the chip will used by many other investigators in the future.

The www.predicta.eu link cited in the introduction, the source of funding for this work, isn't a very helpful source of background information about this project. The manuscript lacks details about methods. What type of printer was used? What substrate were the peptides printed on. What is the concentration of peptide and what buffer?

I feel that the results in the manuscript can't be understood without liberal reference to information and figures in the supplement. Some of the most informative results are in the supplementary figures. Supplementary Table 7 seems to be showing assay reproducibility data but it is not entirely clear just from the table what has been done. How do you know that IgG is lower than IgA? Only that the signal intensities (ISU) are lower for IgG than IgG. It will be helpful to provide more explanation of ISAC and ISU to save the reader time looking up other papers, some not so accessible. Supplementary Figure 3 is referred to and highlighted in the results section, but the important figure needs to be looked up in the supplement. I found it inconvenient to have to go back and forth

between the narrative and the supplement to follow through the results section. Legend to Figure 4 needs more detail.

REPLY: According to reviewer's suggestion, we have included more details about the preparation of the microarray in the revised paper (see lines 348-353). We also explained how the calibration curve was generated and what the ISU means (see lines 353-358). We have also added more details to the legend of Figure 4 (see lines 564-569). Information on how the reproducibility data were generated has already been described in Supplemental Information (see lines 157-169).

Reviewer #3 (Remarks to the Author):

The authors have set out to generate a microarray based serologic test that can discriminate between RV-A,B,C RV species, as culprit species during acute asthma episodes in children. This article builds on previous work by a number of the authors that has shown that a strain specific increase in IgG1 anti-VP-1 antibodies peaks by day 42 post inoculation/infection of adult volunteers (healthy and with mild and moderate asthma). They and others have also shown an association with increased IgG1 antibodies to be seen in those with asthma and to correlate to severity of symptoms. In this study they demonstrate in 120 children with acute asthma a rise in IgG and IgA antibodies that identifies RV species and again show those with the greatest increase in antibodies are associated with more severe clinical disease. There is also an age specific response that increases with age that is not surprising and presumably reflects increasing maturity of the immune response or a cumulative exposure to RV species that increases with time. The authors propose that this serological test can discriminate the RV species responsible for acute asthma in children and has the potential to map RV signatures in larger populations and in different respiratory diseases.

I broadly support their conclusions; though have a number of points that require discussion. There were 32/108 children who were PCR negative for RV during their acute presentation, despite seeing a rise in RV antibodies. This is a large proportion. Given the long lag in antibody rise (42 days), prior exposure to other RV species may be responsible for the rise in antibodies, species that lead to subclinical infection. Alternatively, could there have been cross reactivity with other viruses, especially the related enterovirus family (especially EV-68), that can cause similar presentations to RV?

REPLY: We agree that there is a large proportion of children having an antibody increase and a negative PCR result but we would rather link this fact to the limitations of the use of viral swabs and PCR than to the cross-reactivity with other viruses because we have included on the chip a large number of RV peptides representing the common RV species. In fact, we noticed also a mistake which we have made in Figure 3a during the initial submission. There were positive PCR results in children with negative serology and we have now corrected this mistake in the revised version and mentioned also this possibility. Regarding enterovirus: We also included on the chip an N terminal peptide from EV-68 but have not observed any rise in antibody levels specific for this peptide and the sequence identity to P1 peptide of RV01 was rather low (12.5%).

More detail about this cohort would be required. Clinical severity, age, presence or absence of virus symptoms (granted this is difficult in this age group), time from symptoms onset to presentation with acute asthma. Indeed there are limitations to the use of viral swabs and PCR, poor collection techniques, degradation of specimens with storage or delay in transport, reduced sensitivity in

detection where virus transport media is used. It would argue for the repeatability and better internal consistency of a serological test. While promising the applicability of these results to other age groups and disease states would require validation in these groups and is perhaps somewhat overstated in the manuscript. Despite the preliminary work done in adult volunteers, immune responses to viruses will be quite different in older age groups with both asthma and COPD, virus carriage and shedding is usually less and presentation with an exacerbation may be more delayed.

The manuscript is well written. The statistics used appear appropriate. The figures were of high quality.

REPLY: We completely agree with the reviewer and thank for the fair comments. In fact, we have initiated the work on using the chip for the measurement of RV-specific antibody responses in different age groups and disease states. Clinical severity and age have already been included in Table 1 of the manuscript, whereas information about time from symptoms onset to presentation with acute asthma has unfortunately not been available. Presence or absence of virus symptoms has been added in the revised Table 1.

Minor points:

1. Page 3, line 16, I would regard CDHR3 as at least one of the probable receptors for RV-C, not possible.

REPLY: Following the reviewer's suggestion we have changed the statement in the manuscript (see lines 67-68).

2. Page 4, line 84, ACOS, GINA now recommends the term Asthma COPD overlap, without syndrome and would recommend using this term throughout the manuscript.

REPLY: We thank the reviewer for this information and have now used this new term throughout the manuscript.

Questions for the authors:

1. In those who were PCR negative for RV at presentation, were the differences in age, sex, symptoms of a virus infection, medication use or time from symptom onset to presentation with acute asthma?

REPLY: Following the reviewer's suggestion, we have analyzed children with a positive (n=82) and a negative PCR result (n=26) in regard to differences in age, sex, duration of respiratory symptoms and the use of medication. Children for whom a PCR result could not be obtained were excluded from the analyses. However, no significant differences were observed regarding any of the parameters.

Reviewer's Table 1. Comparison of basic characteristics between children with a positive and a negative PCR result.

Characteristic	Children with a positive PCR result (n=82)	Children with a negative PCR result (n=26)	p value
Age (months)			0.9815
Median	18	18.5	
Range (Min-Max)	6-42	6-39	
Sex			0.8182
Male:female ratio	50:32	15:11	
Male (%)	39	58	
Weeks until follow-up visit			0.8936
Median	11.5*	12*	
Range (Min-Max)	7-25	8-27	
Days with respiratory symptoms (%)			0.5322
Median			
Range (Min-Max)	9.5*	13*	
Days with β_2 -agonists (%)			0.2567
Median			
Range (Min-Max)	24.5*	19*	
* ₁ or ₂ values are missing for this variable;	0-100	0-100	

2. Where other viruses investigated for at the time of presentation? In particular enteroviruses where a cross reactivity may occur?

REPLY: Testing for other respiratory viruses which might have caused wheezing illness in these children had already been done in a previous study of Stenberg-Hammar *et al.* (Subnormal levels of vitamin D are associated with acute wheeze in young children. Stenberg Hammar K, Hedlin G, Konradsen JR, Nordlund B, Kull I, Giske CG, Pedroletti C, Söderhäll C, Melén E. *Acta Paediatr.* 2014 Aug;103(8):856-61). In this study, a molecular diagnostic platform for the rapid detection of 15 respiratory strains and the results was used. See reply to the question of reviewer 1: "In this study, a molecular diagnostic platform for the rapid detection of 15 respiratory strains was used in 118 of the 120 children. The following respiratory viruses were found: Adenovirus: 7 children; Bocavirus: 8 children; Coronavirus: 6 children; Influenza A/B: 1 child; Metapneumovirus: 3 children; Parainfluenzavirus: 4 children: RSV: 22 children. For 108 of the 120 children nasopharyngeal swabs had been available for the RV PCR targeting VP2/4 sequences and positive results for RV were obtained for 82 out of the 108 tested children. Thus RV was by far the most common respiratory virus detected in the children. This was mentioned in the revised supplement section (see lines 105-110). "

Reviewers' Comments:

Reviewer #2 (Remarks to the Author):

The authors have taken by comment into consideration and responded appropriately.

Reviewer #3 (Remarks to the Author):

The authors have adequately addressed the comments that I had posed to them. I accept the clinical data they present is all that is available.

The issue raised by reviewer 1 remains of some concern. If the original kinetic studies that looked at IgG expression were taken at 0, 4, 7 and 42 days there is a long window between day 7 and 42 for another infection to come along and potentially lead to the antibody response seen. Given the problem with so many PCR results not detecting the presence of acute infections more detail is required to see what the kinetics in the change in IgG between day 7 to 42.

Point by point reply

Reviewer #2 (Remarks to the Author):

The authors have taken by comment into consideration and responded appropriately.

REPLY: We thank the reviewer for this positive comment.

Reviewer #3 (Remarks to the Author):

The authors have adequately addressed the comments that I had posed to them. I accept the clinical data they present is all that is available. The issue raised by reviewer 1 remains of some concern. If the original kinetic studies that looked at IgG expression were taken at 0, 4, 7 and 42 days there is a long window between day and 7 and 42 for another infection to come along and potentially lead to the antibody response seen. Given the problem with so many PCR results not detecting the presence of acute infections more detail is required to see what the kinetics in the change in IgG between day 7 to 42.

REPLY: We thank the reviewer for this comment and followed the advice to insert into the discussion a sentence pointing out the limitation of our study “One limitation of our study is however, that we cannot exclude that infections with additional RV strains have occurred in the time window between the first and second blood sampling and thus are responsible for the discrepancy between PCR results and serology.”